# Plasmid-Mediated Spread of Carbapenem Resistance in *Enterobacterales*: A Three-Year Genome-Based Survey

**DOI:** 10.3390/antibiotics13080682

**Published:** 2024-07-23

**Authors:** Yancheng Yao, Can Imirzalioglu, Linda Falgenhauer, Jane Falgenhauer, Petra Heinmüller, Eugen Domann, Trinad Chakraborty

**Affiliations:** 1Institute of Medical Microbiology, Justus Liebig University Giessen, Schubertstrasse 81, 35392 Giessen, Germany; can.imirzalioglu@mikrobio.med.uni-giessen.de (C.I.); linda.falgenhauer@hlfgp.hessen.de (L.F.); jane.c.falgenhauer@mikrobio.med.uni-giessen.de (J.F.); trinad.chakraborty@mikrobio.med.uni-giessen.de (T.C.); 2German Center for Infection Research (DZIF), Partner Site Giessen-Marburg-Langen, 35392 Giessen, Germany; 3Institute for Hygiene and Environmental Medicine, Justus Liebig University Giessen, Schubertstrasse 81, 35392 Giessen, Germany; 4Hessisches Landesamt für Gesundheit und Pflege (HLfGP), Heinrich-Hertz-Strasse 5, 35683 Dillenburg, Germany; petra.heinmueller@hlfgp.hessen.de

**Keywords:** Gram-negative bacteria, carbapenem resistance, WGS, plasmid, surveillance, Germany, implant-associated infections

## Abstract

The worldwide emergence and dissemination of carbapenem-resistant Gram-negative bacteria (CRGNB) is a challenging problem of antimicrobial resistance today. Outbreaks with CRGNB have severe consequences for both the affected healthcare settings as well as the patients with infection. Thus, bloodstream infections caused by metallo-ß-lactamase-producing *Enterobacterales* can often have clinical implications, resulting in high mortality rates due to delays in administering effective treatment and the limited availability of treatment options. The overall threat of CRGNB is substantial because carbapenems are used to treat infections caused by ESBL-producing *Enterobacterales* which also exist with high frequency within the same geographical regions. A genome-based surveillance of 589 CRGNB from 61 hospitals across the federal state Hesse in Germany was implemented using next-generation sequencing (NGS) technology to obtain a high-resolution landscape of carbapenem-resistant isolates over a three-year period (2017–2019). The study examined all reportable CRGNB isolates submitted by participating hospitals. This included isolates carrying known carbapenemases (435) together with carbapenem-resistant non-carbapenemase producers (154). Predominant carbapenemase producers included *Klebsiella pneumoniae*, *Escherichia coli*, *Citrobacter freundii* and *Acinetobacter baumannii*. Over 80% of 375 carbapenem-resistant determinants including KPC-, NDM-, VIM- and OXA-48-like ones detected in 520 *Enterobacterales* were plasmid-encoded, and half of these were dominated by a few incompatibility (Inc) types, viz., IncN, IncL/M, IncFII and IncF(*K*). Our results revealed that plasmids play an extraordinary role in the dissemination of carbapenem resistance in the heterogeneous CRGNB population. The plasmids were also associated with several multispecies dissemination events and local outbreaks throughout the study period, indicating the substantial role of horizontal gene transfer in carbapenemase spread. Furthermore, due to vertical and horizontal plasmid transfer, this can have an impact on implant-associated infections and is therefore important for antibiotic-loaded bone cement and drug-containing devices in orthopedic surgery. Future genomic surveillance projects should increase their focus on plasmid characterization.

## 1. Introduction

The rise of carbapenem-resistant Gram-negative bacteria (CRGNB) represents an increasing threat to patient safety and healthcare systems worldwide [1,2,3,4]. Carbapenemases are the main cause of carbapenem resistance in carbapenemase-producing *Enterobacterales* (CPE). Infections with CPEs are associated with high levels of mortality due to delays in administrating effective treatment and the limited availability of treatment options [5,6]. This requires consistent surveillance, molecular epidemiology, adjusted antibiotic treatment and careful antibiotic stewardship. CPEs generally harbor three classes of carbapenemases: Class A, including KPCs, while VIM and NDM represent class B metallo-β-lactamases (MBLs). Class D includes oxacillinases such as OXA-48 and other related carbapenemases [7,8]. Carbapenem resistance is commonly carried by mobile genetic elements, including plasmids, which harbor different repertoires of carbapenemases and transfer them both within and across multiple species [9,10].

To understand the continuing expansion of carbapenem resistance, enhanced surveillance of CRGNB at local, regional, national and international levels has been recommended [9,11,12]. Surveillance strategies across the EU/EEA countries report widely differing proportions of carbapenem resistance among CPEs [2]. In Germany, continuous increases in CRGNB have been reported since 2009 [2,13], but it is presently not clear as to whether overall trends seen in the national statistics are representative of regional distributions. Since 2011, the state of Hesse was the first federal state to require mandatory notification of CRGNB in Germany. Here, the notification numbers of CRGNB during the study period were 452 in 2017, 433 in 2018 and 413 in 2019 [14,15,16]. Additionally, sporadic outbreaks with strong sub-regional distributions of various CPE species and clones have been reported [17].

To obtain information on sporadic outbreaks and to support local infection control and antibiotic stewardship programs, a genome-based regional CRGNB surveillance program (SurvCARE Hesse) was initiated. The SurvCARE initiative targeted all hospitals geographically distributed across six healthcare districts in the state Hesse. Information from isolates obtained between 2016 and 2019 was combined with their genome-based data to obtain an overview of the overall landscape of CRGNB in Hesse during this period.

## 2. Results

### 2.1. General Characterization of the CRGNB Population

The CRGNB isolates were obtained from 26 (2017), 31 (2018) and 41 institutions (2019), respectively, and represented 25%, 26% and 35% of all hospitals reporting CRGNB isolates to the local public health authorities in the state.

Of the 589 analyzed isolates, 571 were from patients and the other 18 were from hospital environmental samples or sources of unknown origin. Of the patient isolates, 352 (61.6%) were male and 219 (38.4%) were female, and they were mostly from elderly patients over 59 years of age (Appendix A). Most of the patient isolates (86%, n = 492) represented carriage isolates derived from screening samples (rectal swab/stool, skin, groin, nose/throat swab) and approximately 14% (n = 79) were associated with infection (blood culture, urine, invasive sources, patients with symptomatic infections, Appendix A). Overall, 25 different species of CRGNB were detected (Figure 1). A total of 520 (88%) were *Enterobacterales* isolates from 21 different species and 69 (12%) were members of the non-fermenter group, including *Acinetobacter baumannii* (n = 53) and *Pseudomonas aeruginosa* (n = 13).

Based on genomic analyses, at least one known carbapenemase-encoding gene was detected in 435 (74%) CRGNB isolates, including KPC-, NDM-, VIM- and OXA-48-like types and *Acinetobacter* oxacillinases (OXA-23, OXA-72) distributed in 50 hospitals in six healthcare districts of the state (Table 1, Figure 1). In total, KPC-type, OXA-48-type and MBL carbapenemase-producing isolates were found at 27, 28, and 30 participating hospitals across the six healthcare districts. For 154 (26%) CRGNB isolates, no known carbapenemases were detected. There was no gender bias in the proportion of carbapenemase-producing and non-carbapenemase-producing CRGNB.

### 2.2. Population Structure of Escherichia coli

In order to determine the population structures of the main CRGNB species, phylogenetic analyses of sequenced isolates were performed. Figure 2 shows the phylogenetic tree of carbapenemase-resistant *Escherichia coli* isolates, together with the assigned sequence types (MLSTs), the carbapenemase and other beta-lactamase-encoding genes as well as the carbapenemase-bearing plasmid types. The genomes of these 110 sequenced isolates belonged to 40 different STs. The frequency of the individual STs varied from one to thirty isolates. For approximately 81%, viz., 90 genomes in 34 ST types that originated from patients in 20 hospitals, at least one carbapenemase-encoding gene, e.g., KPC-2, OXA-48, OXA-48-like (OXA-244, OXA-181 and OXA-204), NDM-1 or NDM-5, was identified. KPC-2 (n = 21) was found in 11 different STs (ST69, ST131, ST58 and another eight STs). NDM-5 was detected in another 10 STs (mainly ST940, ST1284, ST405 and ST61), while OXA-48 was identified in a further eight STs, most notably ST38, ST354 and ST127. The carbapenemase-positive proportion of human-specific pathogenic *E. coli* ST131 was relatively low (n = 5/10; 50%). In contrast, OXA-244 (a variant of OXA-48) or OXA-48-bearing *E. coli* of ST38, were much more common (n = 25/26; 96%). Carbapenemases of VIM types were not found in the *E. coli* isolate.

### 2.3. Population Structure of Klebsiella pneumoniae

Figure 3 shows the phylogenetic relationships of the 210 carbapenem-resistant *Klebsiella pneumoniae* isolates that belonged to 59 different sequence types (STs). Nearly eighty percent of the 167 isolates that originated from 39 different hospitals and belonged to 44 genome types carried known carbapenemase genes. Ten different carbapenemases were identified in the genomes. The most important carbapenemase types were KPC-3, OXA-48, KPC-3, NDM-1 and OXA-232, which associated with the ST types (ST101, ST307, ST147, ST231, ST512 and ST14) and accounted for the largest proportion. They are associated with diverse clonal outbreaks (either local or between hospitals) as determined by genome-based epidemiological analysis. These include the following: (i) KPC-3 *K. pneumoniae* ST101 (CG43); (ii) NDM-1 *K. pneumoniae* ST147; (iii) OXA-232 *K. pneumoniae* ST231; (iv) OXA-232 *K. pneumoniae* ST2096 (CG231); and (v) KPC-3 *K. pneumoniae* ST512 (CG258). Within the *Klebsiella pneumoniae* carbapenemase (KPC)-producing *K. pneumoniae* isolates, ST512 was more dominant than ST258. A peculiarity not previously described in Germany is that *K. pneumoniae* ST307 is linked to various carbapenemases such as KPC-3, OXA-48, NAM-1 and OXA-181, and it appears to be a new proliferating regional sequence type, which has also been described simultaneously in Italy and France. In addition, numerous STs are involved in plasmid-mediated KPC-2 transmission, which occurs both intra- and inter-species (see below for details).

### 2.4. Population Structure of Citrobacter Species

Although *Citrobacter* species can colonize the human gut, this occurs far less frequently than by *E. coli* or *K. pneumoniae*. *Citrobacter* species were only rarely described as a pathogen of human infections and are more prevalent in the environment. In our study, we observed an abundance and marked increase in the occurrence of carbapenem-resistant *Citrobacter* spp. including *C. freundii*, *C. braakii*, *C. koseri* and *C. portucalensis* (Figure 4). The species *C. freundii* was the third most common clinical species (n = 50) detected among the family *Enterobacterales*. It was associated with the highest numbers of carbapenemases and geographically distributed to 15 hospitals within the region. Phylogenetic analysis revealed a high genetic diversity of the *C. freundii* isolates assigned to 30 different sequence types (Figure 4). Over 94% (50/53) of them harbored carbapenemase genes. Many carbapenemase types, KPC-2, KPC-3, OXA-48, OXA-162, VIM and NDM, were identified. Isolates of ST18, ST19, ST22 and ST98 each occurred several times, while the other STs were each detected only once during the entire study period. The carbapenemases of the *C. freundii* isolates were found in diverse STs, i.e., KPC-2 in eleven STs and OXA-48 in nine STs.

### 2.5. Population Structure of Enterobacter Species

The carbapenem-resistant isolates of the *Enterobacter cloacae complex* were distantly related to one another (Figure 5). Phylogenomic analysis classified isolates as *Enterobacter asburiae*, *E. cloacae*, *E. roggenkampii* and *E. xiangfangensis*. They were members of different sequence types including several hitherto unknown ST types. A total of 86% of them contained the ACT allele *ampC* and 79%, the fosfomycin-resistant gene *fosA*. Carbapenemase-encoding genes were only identified in about one third of the isolates, often co-existing with several other beta-lactamase genes such as *bla*_CTX-M-15_ or *bla*_CTX-M-9_, *bla*_OXA-1_ and *bla*_TEM-1B_. Overall, KPC-2 in the *Enterobacter* spp. isolates accounted for more than half of all carbapenemase genes detected. The identified *bla*_KPC-2_ genes were almost exclusively all located on an IncN plasmid (pMLST15) (Figure 5).

### 2.6. Population Structure of Acinetobacter baumannii

*Acinetobacter baumannii* is a known and feared pathogen of nosocomial infections and outbreaks, especially in intensive care units and in immunocompromised patients. For each of the 54 isolates (from 13 different hospitals and 24 laboratories), at least one carbapenemase-encoding gene was identified. *A. baumannii* isolates are members of 28 different ST types according to the Oxford scheme or 15 STs according to the Pasteur scheme, and they belong mainly to the ST2 and ST636 groups (Figure 6). The most common carbapenemase type detected was *Acinetobacter* oxacillinase OXA-23 (64%), followed by OXA-72 (27%), NDM-1 (5%), NDM-5 (2%) and GES-11 (2%). Two isolates each harbored two carbapenemases, of which one contained OXA-23 and NDM-5 and the other OXA-58 together with NDM-1 (Appendix A). The OXA-23-producing *A. baumannii* isolates were often members of ST2 while the OXA-72 (OXA-42/24-like) producers were members of ST636.

### 2.7. Genetic Characteristics of Remaining Species

Figure 7 depicts the isolates of the remaining 14 carbapenem-resistant bacterial species. They comprised *K. aerogenes*, *K. oxytoca*, *K. michiganensis*, *K. variicola*, *Serratia marcescens*, *Proteus mirabilis*, *Providencia rettgeri*, *Morganella morganii*, *Cedecea lapagei*, *Raoultella planticola* and *R. ornithinolytica* together with *Acinetobacter pittii*, *Pseudomonas aeruginosa* and *P. stutzeri*. Phylogenetic analysis revealed clonal relations present in isolates among *S. marcescens* (VIM-1-positive), *K. michiganensis* (VIM-1-positive) or *K. oxytoca* (non-known carbapenemase genes). Isolates of *K. aerogenes* species were genetically very divergent and did not harbor any currently known carbapenemase gene. The thirteen *Pseudomonas aeruginosa* isolates belonging to nine different sequence types from seven hospitals indicated that seven of them contained carbapenemase-encoding genes for Verona integron-encoded metallo-beta-lactamases VIM-2 and VIM-4 in their genome. To summarize, carbapenemase-producing CRGNB were found in 16 *Enterobacterales* species and two non-fermenter species, *A. baumannii* and *P. aeruginosa*. *K. pneumoniae*, *E. coli* and *C. freundii* were the three most frequent *Enterobacterales* species detected and accounted for 81% of all isolates.

### 2.8. Distribution of Carbapenemases Detected

KPC-2 was the most frequent carbapenemase type detected (n = 85; 20%) and occurred in multiple *Enterobacterales* species (n = 11) and across various STs (Figure 1C, D, Appendix A). In contrast, KPC-3 represented 11% (n = 47) of all carbapenemases, and it was restricted to a few STs of *K. pneumoniae* (87%) (ST101, 32%; ST512, 23%; ST307, 17%) and *C. freundii* (ST18, 9%).

OXA-48 accounted for 18% (n = 77) of all carbapenemases and was found frequently in *K. pneumoniae*, *E. coli* and *C. freundii*, and it was represented by seventeen, nine and eight STs, respectively (Figure 1C, D, Appendix A). OXA-48-like carbapenemases (OXA-244, OXA-232, OXA-181, OXA-162, OXA-143 and OXA-204) comprised 15% (n = 65) of all carbapenemases detected. Of these, 21 were *E. coli* ST38 OXA-244-producing isolates (Figure 2). OXA-232 (n = 18) was found in *K. pneumoniae* ST232 and ST2096 (Figure 3).

*Klebsiella* spp. was the most prevalent MBL producer (42%, 52/123), followed by *Serratia marcescens* (14%, 17/123), *C. freundii* (9%, 11/123) and *P. aeruginosa* (6%, 7/123). The distribution of MBL types varied significantly (Figure 1D). We found 28% (17/61) of VIM in *S. marcescens* and 54% (34/63) of NDM in *K. pneumoniae*.

### 2.9. Temporal Changes

To examine changes over time, we compared the proportional distribution of the main species with the type of carbapenemases detected between 2017 and 2019. Data collected before 2017 were excluded from this analysis as this was only derived from isolates obtained in the last three months of 2016.

Apart from an increase in absolute numbers from 2017 to 2019, we observed a significant percentile increase in carbapenemase-encoding *E. coli* isolates in 2019 and a significant decrease in *A. baumannii* in 2018 (Figure 8). For the remaining species, no substantial changes were apparent (Table 1 and Appendix A). However, significant changes in several carbapenemase types occurred within the study period (Figure 8). The biggest increases were in OXA-48 and OXA-48-like carbapenemases in 2018 and 2019, associated with *K. pneumoniae*, *E. coli* and *C. freundii*, respectively (Table 1). This was also the case with NDM carbapenemases which showed high annual increases of >51% for 2018 and >85% for 2019 (Appendix A) [18]. While the number of isolates harboring a KPC-2-type carbapenemase steadily increased between 2017 and 2019, those harboring KPC-3 carbapenemases peaked in 2017 and declined significantly in 2018 and 2019.

### 2.10. Emergence of Isolates Harboring More than One Carbapenemase

Eighteen isolates of different species each harbored two different classes of carbapenemases (Appendix A). Most of these isolates were *K. pneumoniae*. Compared to the total number of CRGNB isolates, the proportion of isolates carrying two classes of carbapenemases comprised 0.8% in 2017, 3.5% in 2018 and rose to 4.4% in 2019, indicating an upward trend. Five isolates carried VIM-1 with either OXA-162 (*S. marcescens,* n = 3) or KPC-2 (*E. cloacae* novel ST, *C. freundii* novel ST). Five other isolates harbored NDM-5 with either OXA-181 (*K. pneumoniae* ST147 n = 2; *E. coli* ST2851), OXA-48 (*K. pneumoniae* ST383) or OXA-23 (*A. baumannii* ST684). Two *A. baumannii* isolates harbored a combination of NDM-1 with either OXA-23 or OXA-68. One *K. pneumoniae* isolate harbored a combination of NDM-1 with OXA-232. The remaining isolate was *E. xiangfangensis* ST88, which carried both KPC-2 and OXA-48 carbapenemases.

### 2.11. Characterization of Plasmids in Carbapenemase-Producing CRGNB

Genome-based analysis revealed that more than 97% of carbapenemase-encoding *Enterobacterales* isolates harbored plasmids. A total of forty different plasmid types were identified in carbapenemase-producing *Enterobacterales* isolates (Appendix A). The most frequently identified plasmid Inc groups were IncFIB, IncFIB(*K*), IncF(*K*), IncFII, IncHI1, IncN, IncL/M, IncR and IncX3, carrying different carbapenemase genes (Appendix A). As reported previously [10,19], these plasmids differed in their host range. For instance, *bla*_KPC-2_-encoding IncNs (pMLST-15) were identified in more than 10 different species and distributed in all healthcare districts, while the *bla*_KPC-3_-encoding IncF[*K1:A-B-*] plasmid was detected only in *K. pneumoniae* in two healthcare areas (Figure 9, Appendix A). The *bla*_KPC-2_-encoding IncN (pMLST-15) increased continuously from 2017 to 2019 in *K. pneumoniae* and *E. coli* (Appendix A). IncFIB-type plasmids were frequently identified in both *K. pneumoniae* and *E. coli*, but IncFIB(*K*) was found more frequently in *K. pneumoniae* and IncFII in *E. coli* (Appendix A). For the *A. baumannii* ST636 clone (n = 12), OXA-72 was located on a Rep-3-associated GR20-type plasmid.

We found that IncN plasmids (ST-7, ST-7-like) harbored 52% of VIM-1 (n = 52), while 51% of KPC-3 (n = 47) isolates carried the gene on IncF(*K*) plasmids. In addition, 56% of OXA-48 (n = 77) were present on IncL/M(pOXA-48) plasmids, with 18% of all NDM-5 (n = 28) isolates carrying the gene on a type IncFII plasmid (Figure 9).

By combining data on genetic typing, carbapenemase-type, MLST, the respective plasmid incompatibility groups and/or pMLST-type together with antibiotic-resistant genes, 38 clonal clusters in 40 hospitals were detected (Appendix A). While some of these clonal clusters (8, 32) were clearly local and restricted to a single health district, other clonal clusters (20, 21) were more widespread and present in geographically separated districts. In total, they accounted for 41% (243/589) of all investigated isolates.

### 2.12. Genetic Characteristics of Non-Carbapenemase-Producing CRGNB

One hundred and fifty-four (26%) CRGNB did not harbor any known carbapenemases. They belonged to 18 species and ~91% were isolates of the species *K. pneumoniae* (n = 41), *K. aerogenes* (n = 34), *E. coli* (n = 19), *Enterobacter cloacae* (n = 15), *Enterobacter xiangfangensis* (n = 11), *K. oxytoca* (n = 11), *P. aeruginosa* (n = 6), and *E. kobei* (n = 3). The remaining non-carbapenemase-producing CRGNB isolates (n = 14) identified were *Acinetobacter pittii*, *Cedecea lapagei*, *Citrobacter freundii*, *Citrobacter braakii*, *Klebsiella variicola*, *Morganella morganii*, *Proteus mirabilis*, *Providencia rettgeri, Providencia stutzeri* and *Serratia marcescens*. It was noteworthy that seven species, *A. pittii*, *C. lapagei*, *E. kobei, M. morganii*, *P. mirabilis*, *P. rettgeri and P. stutzeri*, harbored exclusively non-carbapenemase-producing isolates (n = 12). The genome types identified are shown in Appendix A.

*Enterobacter* spp. was a relevant CRGNB in our study; however, 67% of their isolates did not carry any known carbapenemase (Figure 5). A high proportion of non-carbapenemase-mediated carbapenem resistance was also found in *K. aerogenes* (94%), *K. oxytoca* (85%) and *P. aeruginosa* (46%) (Figure 7), while this proportion was relatively low in isolates of *K. pneumoniae* (20%), *E. coli* (19%) and *C. freundii* (3%). An increased proportion of isolates of *P. aeruginosa* and *K. aerogenes* without known carbapenemases was detected in 2018 and 2019 (Appendix A). For the remaining species, no distinct trend was apparent.

Ten (24%) of the non-carbapenemase-producing *K. pneumoniae* isolates carried at least one class-A ESBL gene, eight (20%) harbored two ESBL genes and a further 23 isolates (56%) had more than two genes (Appendix A). Genetic alterations such as deletions, substitutions or premature stop codons were identified in the genes encoding *ompK*35 (12%) and *ompK*36 (more than 24%) genes. For the non-carbapenemase-producing *K. aerogenes* mutations in the *ampC*, *ampD* and *ampR* and *ompK*36 genes were identified.

## 3. Discussion

Bacteria of the family *Enterobacteriaceae* such as *E. coli* or *K. pneumoniae* are indigenous members of the human intestinal flora and important causes of healthcare-associated infections. Increases in the local consumption of carbapenems has been associated with the expansion of CPEs and particularly that of carbapenem-resistant *K. pneumoniae*, which have a high potential of causing outbreaks in healthcare settings [20,21]. Our results document the predominance of *K. pneumoniae* as a carrier of all classes of carbapenemases, reflecting its role as a resistance reservoir in healthcare settings and a major disseminator of resistance determinants within the region. We detected significant temporal changes in the epidemiology of several NDM and OXA-48 variants with a clear upward trend associated with several clonal outbreaks in 2018 (OXA-232 *K. pneumoniae* ST231) and 2019 (NDM-1 *K. pneumoniae* ST147).

Compared to studies on carbapenemase-producing *Enterobacterales* performed in other countries during the same study period (ATLAS study [22], many different countries, Peru [23], Australia [24], Japan [25]), our study revealed differing carbapenemase prevalences. In our study, the overall percentage of OXA-48(-like)-positive *Enterobacterales* (33%) was far higher than in all the above-mentioned studies (ATLAS study 2.6% OXA-48(-like), Peru 0%, Australia 15% OXAs, Japan < 1%). IMP-like carbapenemases were not detected in our study, while they were the most common carbapenemases in the Japanese study and the second most common carbapenemases in the Australian study. The prevalence of NDM producers in our study (19.9%) was far lower than in the study performed in Peru (90.5%), about a half of the prevalence of NDM producers in the study performed in Australia (38%) and around three times higher than in the Japanese study (7.4%).

Within the species *K. pneumoniae*, the prevalence of KPC (40.9%) and OXA-48-type carbapenemases (35.7%) seen here was comparable to that of the EuSCAPE study which covered many European countries [11]. The prevalence of NDM was higher in our study (19.9%) and conversely lower for VIM (3.5%). Also, in this study, the dominant STs were ST307 for all carbapenemases, ST147 for NDM, ST101, ST512 and ST14 for KPC and OXA-48, and ST231 for OXA-233. The predominant STs previously reported in Europe were ST258, ST15, ST101 and ST11 [11]. A study in Spain on carbapenemase-producing Gram-negative bacteria in Andalusia also reported a different distribution of associated carbapenemase types [12]. An outbreak of carbapenem-resistant *K. pneumoniae* ST307 affecting 17 patients occurred in 2019 in the northeastern region of Germany, indicating the supra-regional emergence of carbapenem-resistant *K. pneumoniae* ST307 [21]. Compared to the situation of 2008 to 2014 in Germany [26], where KPC (56%), OXA-48 (39%) and NDM (5%) were observed, our study clearly shows differences in KPC- and MBL-type carbapenemases as compared to other regions in Germany. This might also be related to different sampling periods.

The species *Citrobacter freundii* is also a significant carrier of different classes of carbapenemases, and clonally related isolates were obtained from various hospitals. Previous reports documented the repeated isolation of carbapenem-resistant *Citrobacter* species in Europe [27] and it appears to be emerging as an important reservoir of many types of carbapenemases [28]. On the other hand, *Serratia marcescens* isolates harboring the metallo-β-lactamase VIM-1 were associated with a large outbreak at a single institution. A clone of *K*. *michiganensis* carrying the VIM-1 gene was detected in four neighboring hospitals in a single health district.

Carbapenem-resistant *E. coli* associated with healthcare-associated infections pose a greater risk because of their ability to spread into the community. Although the proportions of *E. coli* harboring KPC-2 and OXA-48 remained high in the period surveyed, we noted the emergence of new clones of *E. coli* expressing NDM-5 and OXA-244 in this regional study, particularly since 2017 (Figure 8, Appendix A). Thus, OXA-244-producing clonal lineages of ST38 that are members of the *E. coli* phylogenetic group D commonly associated with extraintestinal infections appeared repeatedly at different hospitals, albeit in only two health districts. Due to the inherent difficulty in detecting isolates expressing the OXA-244 allele, it is likely that this group of extraintestinal pathogenic *E. coli* (ExPEC) was underreported in this study. Also noteworthy is the detection of distinct *E. coli* clones (ST167, ST405 and ST361), which are successful epidemic clones associated with ExPEC infections and harbor the class B metallo-β-lactamase NDM-5. The increase in NDM-5 was associated with several IncFII or IncX3 plasmids in 2019 [29].

This study revealed an extraordinary role of specific extrachromosomal genetic elements in CRGNB. Detailed analysis indicated that only seven plasmids accounted for almost 45% (173/384) of all carbapenemases detected in *Enterobacterales* (Figure 9). Thus, 85% (74/85) of the KPC-2-positive isolates harbored a near-identical *bla*_KPC-2_-encoding IncN-plasmid of pMLST-15 (Appendix A) with a low copy number (~2) and a median size of around 78 kb. This highly promiscuous plasmid was present in 10 different *Enterobacterales* species from 64 patients in 17 hospitals (Figure 9). The highly conserved plasmid backbone includes a specific and unique *bla*_KPC-2_ genetic element that was only present in plasmids detected in England and Germany [30,31]. This finding indicates that the spread of KPC-2 is mediated almost exclusively by a single specific plasmid that has persisted over a long period in this region.

KPC-3 is almost exclusively associated with *K. pneumoniae* in particular, with the international high-risk clones ST101 (CG43) and ST512 (CG258), which are endemic in other regions in Germany and countries in Europe [12,20,26]. Nevertheless, the high numbers of KPC-3-expressing *K. pneumoniae* isolates on highly related IncF plasmids reflects a truly regional distribution, as this allele of Class A carbapenemases has only been reported nationwide very rarely.

The VIM-1 allele was predominantly observed in the *K. michiganensis* clones ST213 and ST180, ST45 of *K. pneumoniae*, ST19 of *C. freundii* and *S. marcescens*, but associated with the plasmid types IncN pMLST-7 and pMLST-7-like. For OXA-48 carbapenemases, 53% (41/77) were encoded on IncL/M replicons, further underlining the importance of plasmids in the dissemination of these enzymes [26,32]. Our results showed that IncL/M (pOXA-48) and IncL/M (pMU407) were highly prevalent in the OXA-48-harboring isolates of all species except for *E. coli*.

In *A. baumannii*, OXA-23 and OXA-72 were the most common carbapenemases, with ST2 being the predominant clone, as described in previous studies [33,34]. We detected a small increase in *A. baumannii* isolates harboring a combination of OXA-23/OXA-58 and NDM carbapenemases, which is in agreement with previous studies [35]. Our findings indicate that the carbapenemase-producing *A. baumannii* isolates belonged almost exclusively to the international clone II, consistent with the global view [36,37].

For those isolates that did not harbor any known carbapenemases, we found a rate of 20% for *K. pneumoniae* and 19% for *E. coli*. This compares with previous data from a European-wide study of 29.3% (353/1203) for *K. pneumoniae* and 60.3% (117/194) for *E. coli*, as in [11]. Non-enzymatic carbapenem resistance mechanisms, which were observed in 26% of our isolates overall, often include the loss of expression of porin-encoding genes, mutations in chromosomally encoded porin genes and overexpression of efflux pumps as well as of beta-lactamases, particularly ESBLs [38,39]. Indeed, our analysis revealed genetic alterations of *ompK*36 in non-carbapenemase-producing *K. pneumoniae* isolates and of *ampC*, *ampD* and *ampR* and *omp*36 in non-carbapenemase-producing CR *K. aerogenes*. Evidence on carbapenemase resistance due to gene alterations of *K. aerogenes* has been reported previously [40].

In summary, this study highlights the presence of dominant clones such as OXA-244-producing *E. coli*-ST38, OXA-232 *Kpn-*ST231, OXA-48 *E. coli*-ST38 and *Kpn*-ST307, KPC-3 *Kpn*-ST101, *Kpn*-ST307 and *Kpn*-ST512, as well as NDM-1 *Kpn*-ST147 over a period of three years in the state of Hesse. We also documented the prevalence of successful, local and super-regionally emerging plasmids encoding KPC-2, VIM-1 and NDM-5 carbapenemases. As plasmids are major drivers in the case of multispecies carbapenemase outbreaks, efforts must be undertaken in genome-based studies to identify and classify these entities [41]. This can have an impact on implant-associated infections and is therefore important for antibiotic-loaded bone cement and drug-containing devices in orthopedic surgery. In the future, more efforts should be made to investigate related plasmids in epidemiological WGS studies, especially for multispecies plasmid-based carbapenemase outbreaks that would not be detected and analyzed otherwise.

**Limitations:** There are several limitations to this work. Despite regular meeting events and symposia, an average of 34% of the hospitals participated in this study between 2017 and 2019. An extension of such activities to also detect ongoing outbreaks directly would have been desirable in order to provide close to real-time outbreak analysis. Nevertheless, the need to facilitate the accessibility and literacy of WGS-based interpretations in public institutions remains high. We note that the mechanisms of non-carbapenemase-mediated carbapenem non-susceptibility in Gram-negative bacteria described here are somewhat speculative, and a combination of genomic analysis and experimental validation is needed to eventually detect hitherto unknown carbapenemases and their attendant resistance mechanisms. As we collected all CRGNB regardless of species, clones or resistance mechanisms, this allowed us to determine the diversity of resistance mechanisms involved and to identify those cases where molecular mechanisms were initially unrecognizable (e.g., detection of isolates with no known carbapenemases). Furthermore, sampling was restricted locally to the state Hesse in Germany. We believe that an expansion of such studies would provide surveillance data with actionable information for the control of the emergence and spread of CRNGB locally and regionally.

**Conclusions.** Plasmids can be passed from species to species and even across genera among bacteria. This ability to transfer plasmids vertically and horizontally results in a kind of snowball system that leads to a rapid and cross-species spread of resistance.

It is therefore essential to ensure very good hygienic measures to prevent the spread of CRNGB in hospitals as far as possible. This requires very good surveillance, including molecular epidemiology, and the consistent application of antibiotic stewardship to remove the selective pressure on bacteria.

Hospitals should identify high-risk clones and plasmids (superspreaders) in order to enable targeted surveillance and to adjust the usage of antibiotics. Complex genomic analyses represent one of the most powerful tools for explaining the spread of pathogens and antibiotic resistance and are an excellent basis for combating health threats.

## 4. Materials and Methods

### 4.1. Study Setting and Isolate Collection

Hesse is a German federal state with ~6.2 million inhabitants and an area of 21,114 km^2^. The regional healthcare system is subdivided in six healthcare districts and comprises 127 hospitals. During the study period, 61/127 hospitals voluntarily participated in the SurvCARE initiative (Appendix A). A total of 621 isolates were recruited and 589 CRGNB isolates detected using the German Commission for Hospital Hygiene and Infection Prevention (KRINKO) screening criteria, which were submitted by contributing diagnostic laboratories for further investigation [42].

### 4.2. Sample Size

To estimate the statistical power of our study, we used data on CPE prevalence (*p* = 45.2% at overage) from the National Reference Center for Gram-negative bacteria at the Robert Koch Institute in Germany between 2017 and 2019 [14,15,16]. Based on these data, we expected to achieve a statistical power of 95% with a significance level (alpha) of 0.05 for a sample of 381 *Enterobacterales* isolates. With a total sample size of 520 *Enterobacterales* in our study, a coverage rate of 137% was reached. Analogously, for *Acinetobacter baumannii* based on *p* = 96.4%, a coverage rate of 103% was achieved with a sample size of 55 *A. baumannii* isolates. However, a required sample of 279 *Pseudomonas aeruginosa* was not achieved by a lot.

The formula was determined as the following:n = (Za/2)^2^ P (1 − p)/d^2^
where
n = required sample size.Z = standard normal distribution value at 95% confidence level of a ((Za/2) = 1.96).P = proportion of carbapenemase-producing isolates.d = margin of error = 5%.

### 4.3. Antimicrobial Susceptibility Testing

The antimicrobial susceptibility results obtained in the respective hospitals were confirmed centrally using the VITEK^®^ 2 system (bioMérieux, Nürtingen, Germany) and interpreted following EUCAST guidelines. Taxonomy was confirmed using MALDI-TOF-MS (bioMérieux).

### 4.4. Whole-Genome Sequencing

For isolates that were not susceptible to at least one carbapenem, DNA was isolated from overnight cultures using a PureLink Genomic DNA kit (Invitrogen/ThermoFischer, Darmstadt, Germany). Short-read sequencing libraries were prepared using a Nextera XT kit (Illumina, The Netherlands) and sequenced on MiSeq/NextSeq 500 sequencing machines (read length 2 × 300 nt or 2 × 150 nt). Long-read sequencing libraries were prepared using a native barcoding kit (EXP-NBD103; Oxford Nanopore Technologies, Oxford, UK) and 1D chemistry (SQK-LSK108; Oxford Nanopore Technologies, Oxford, UK). Sequencing was performed on a MiION sequencer (Oxford Nanopore Technologies, Oxford, UK) using a SpotON ML R9 Version flow cell (FLO-MIN106; Oxford Nanopore Technologies, Oxford, UK). Post-sequencing quality control and assembly (short-read or hybrid) was performed using the ASA^3^P pipeline [43], and if needed, the CLC Genomics Workbench v.10.1.0 (Qiagen, Aarhus, Denmark) was also used. The average read length was 185 nt and the average coverage 106x (Appendix A).

### 4.5. Genomic Analysis

Identification of the chromosomal multi-locus sequence types (MLSTs), plasmid incompatibility (Inc) groups, plasmid MLSTs (pMLSTs) as well as acquired antibiotic resistance genes was performed using the Center for Genomic Epidemiology platform (http://www.genomicepidemiology.org/services, accessed on 1 June 2024) and the PubMLST database (https://pubmlst.org, accessed on 1 June 2024; https://bigsdb.pasteur.fr/cgi-bin/bigsdb/bigsdb.pl?db=pubmlst_klebsiella_seqdef&page=sequenceQuery, accessed on 1 June 2024). For *Enterobacter* spp., type strains from the DSMZ, German Collection of Microorganisms and Cell Cultures, were used as a reference for the identification of the *Enterobacter cloacae complex* species.

Phylogenetic comparative genomics was performed when an outbreak or transmission was suspected. MAUVE v.2.3.1 was used for comparative genomic alignments [44]. To identify close genetic relatedness (clonality), single nucleotide polymorphism (SNP)-based phylogenetic analysis was performed using Harvest Suite (ParSNP v.1.2) [45].

For the characterization of carbapenemase-encoding plasmids, specific contigs were analyzed in-depth using read mapping and blastN/P against reference plasmids. The plasmid copy number was calculated by dividing the average coverage of genes defined by pMLST with that of the chromosome.

Genes encoding inducible AmpC cephalosporinase (*ampC*, *ampD*, *ampR*) and outer membrane porins (*omp36*) of the carbapenem-susceptible isolate *K. aerogenes* NCTC10336 and *ompK35* and *ompK36* (Accession Numbers AJ011501 and Z33506) of *K. pneumoniae* were used as references to identify alterations in the non-carbapenemase-producing isolates as previously described [39].

### 4.6. Statistical Tests

General descriptive statistics such as percentages were used. A chi-square test was used for testing of significant differences.

### 4.7. Ethical Approval

Ethical approval was sought at the Ethics Committee of the State Medical Association of Hesse in Frankfurt/Main. The Committee decided on 24 January 2018 that ethical approval of the project was not necessary, as for this study, patient data were rendered anonymous.

## Figures and Tables

**Figure 1 antibiotics-13-00682-f001:**
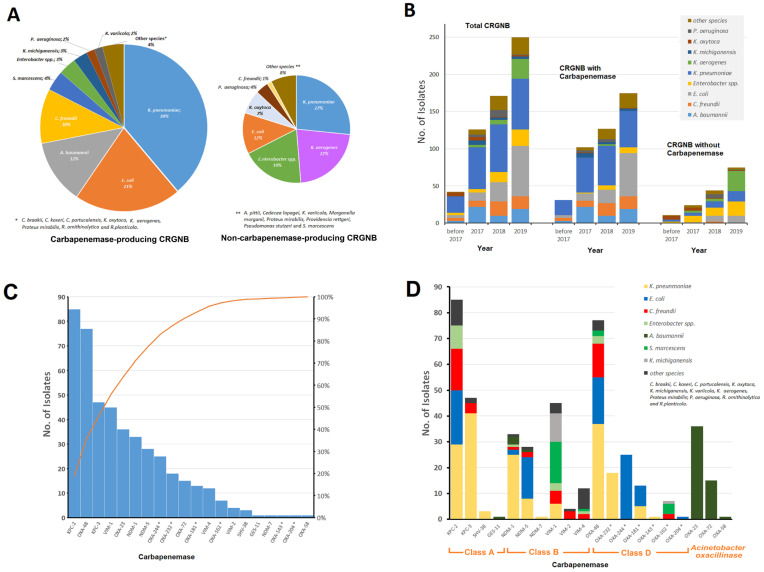
Overview of the CRGNB studied from Hesse, 2016–2019. (**A**) Percentage distribution of carbapenemase-producing (CP) and non-carbapenemase-producing (NCP) CRGBN; (**B**) Year-wise species distribution of all carbapenem-resistant bacteria, CP and NCP-CRGNB; (**C**) Numbers of identified carbapenemase types; (**D**) Carbapenemase distribution according to the species. * = OXA-48-like.

**Figure 2 antibiotics-13-00682-f002:**
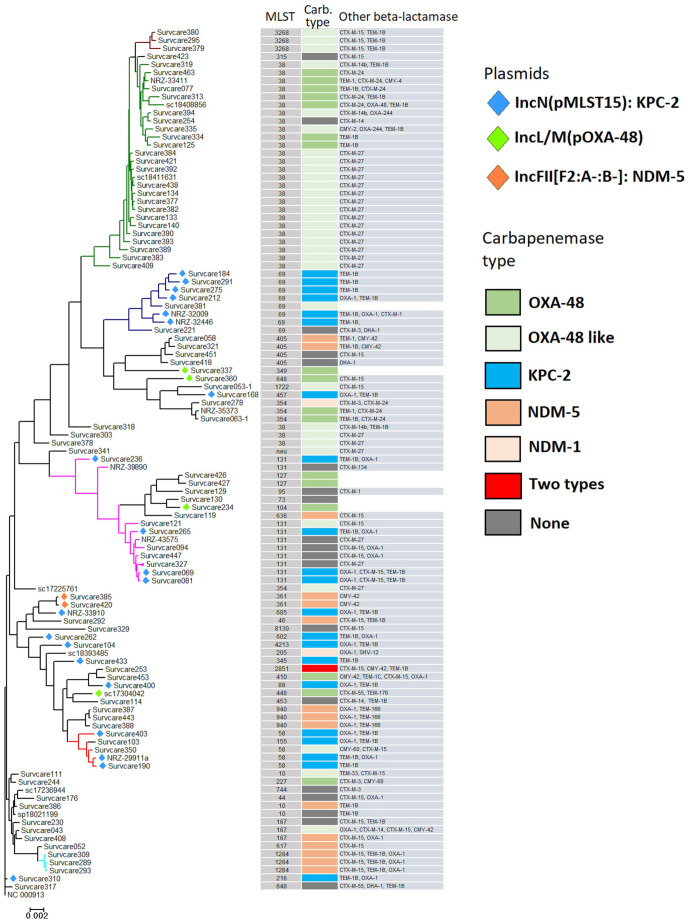
Phylogenetic tree of the carbapenem-resistant *Escherichia coli* isolates. The predicted plasmid types that bore KPC-2, OXA-48 or NDM-5 are indicated directly on the tree branches. MLST types, carbapenemase types and genes encoding other β-lactamases are shown. The different color lines indicate the predominant ST-types: ST38 (green), ST69 (blue), ST131 (magenta), and ST58 (red). The remaining ST’s are shown in black.

**Figure 3 antibiotics-13-00682-f003:**
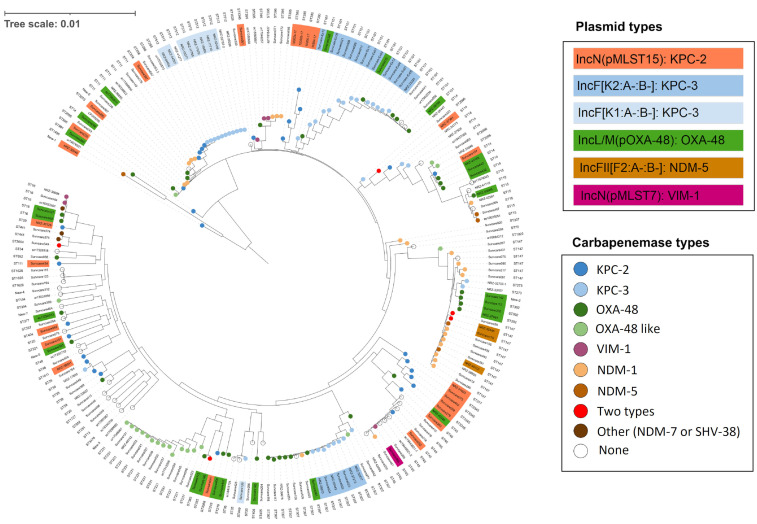
Phylogenetic tree of the carbapenem-resistant *Klebsiella pneumoniae* isolates and the identified carbapenemase-encoding plasmids. Clonal clades belonging to KPC-3-positive ST512, ST101 and ST307, NDM-1-carrying ST147 and OXA-232-encoding ST231 were present. KPC-2 and OXA-48 were distributed broadly in 30 and 18 different ST types, respectively. A total of 80% of KPC-2 genes were predicted to be located on an IncN (pMLST-15) plasmid and 67% of the OXA-48 genes were carried by the plasmid of the IncL/M(pOXA-48) group.

**Figure 4 antibiotics-13-00682-f004:**
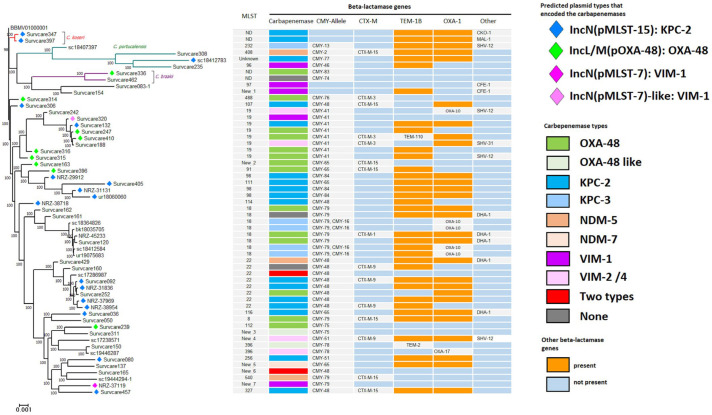
Phylogenetic tree and genomic characterization of the *Citrobacter* spp. The identified carbapenemase-encoding plasmid types are indicated on the tree. Genomes of *Citrobacter freundii* have a large diversity. MLST types, carbapenemase types and other beta-lactamase genes are shown. *Citrobacter* species other as “*freundii*” are indicated: *C. koseri* (red), *C. portucalensis* (green), *C. braakii* (magenta).

**Figure 5 antibiotics-13-00682-f005:**
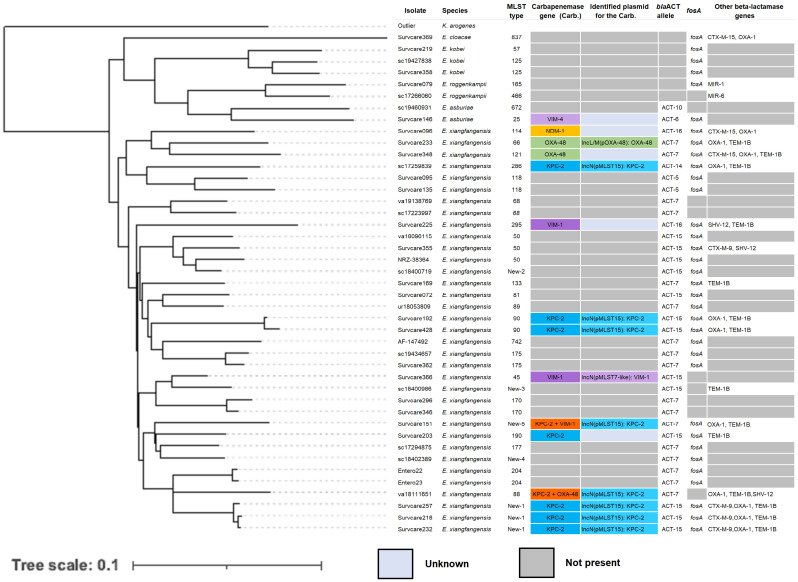
Phylogenetic tree of the carbapenem-resistant *Enterobacter* spp. based on whole genome alignment (MAUVE). MLST types and carbapenemase types as well as other beta-lactamase genes identified are indicated.

**Figure 6 antibiotics-13-00682-f006:**
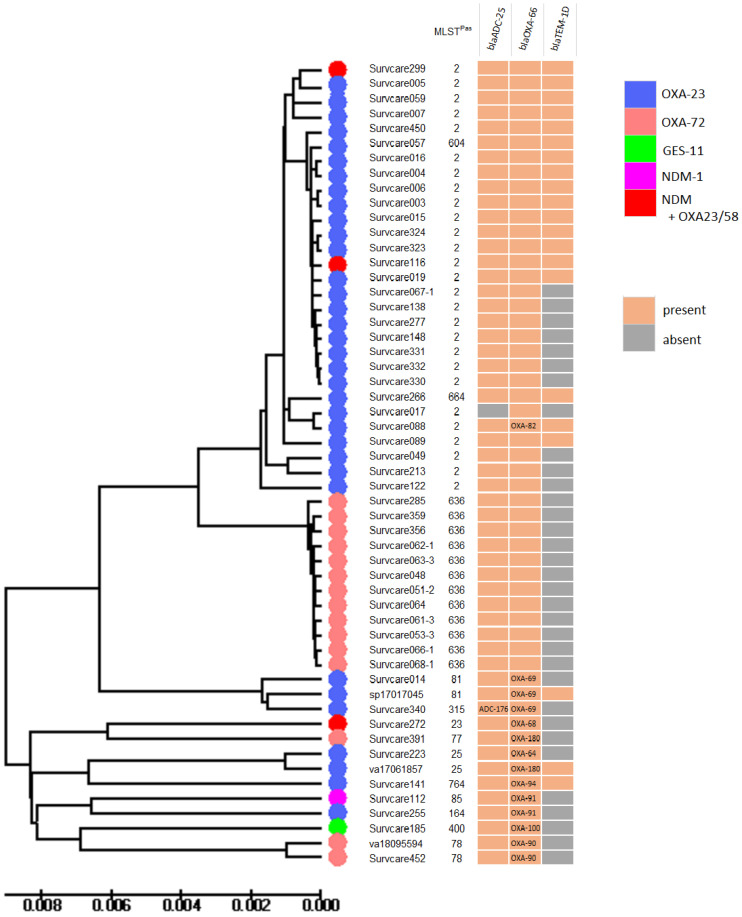
Phylogenetic tree of the *Acinetobacter baumannii* isolates. OXA-23 and OXA-72 were the most predominant carbapenemase types and associated with genome types ST2 and ST636 according to the Pasteur-MLST scheme.

**Figure 7 antibiotics-13-00682-f007:**
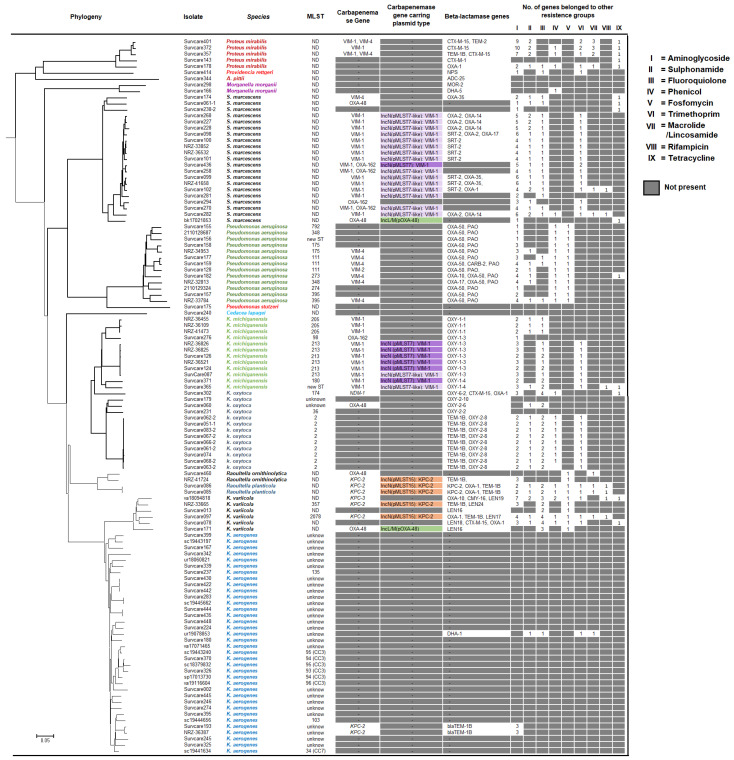
Phylogenetic tree based on whole genome sequence alignment (MAUVE) and the genetic characteristics of all remaining species except *E. coli*, *K. pneumoniae*, *Acinetobacter baumannii*, *Citrobacter* spp. and *Enterobacter* spp.

**Figure 8 antibiotics-13-00682-f008:**
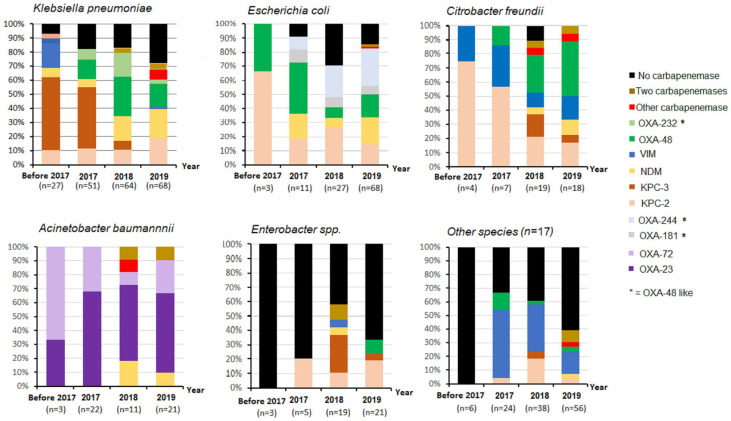
Relative distribution of carbapenemase types within the predominant species *Klebsiella pneumoniae*, *Escherichia coli*, *Citrobacter freundii*, *Acinetobacter baumannii* and *Enterobacter* spp. according to the study periods.

**Figure 9 antibiotics-13-00682-f009:**
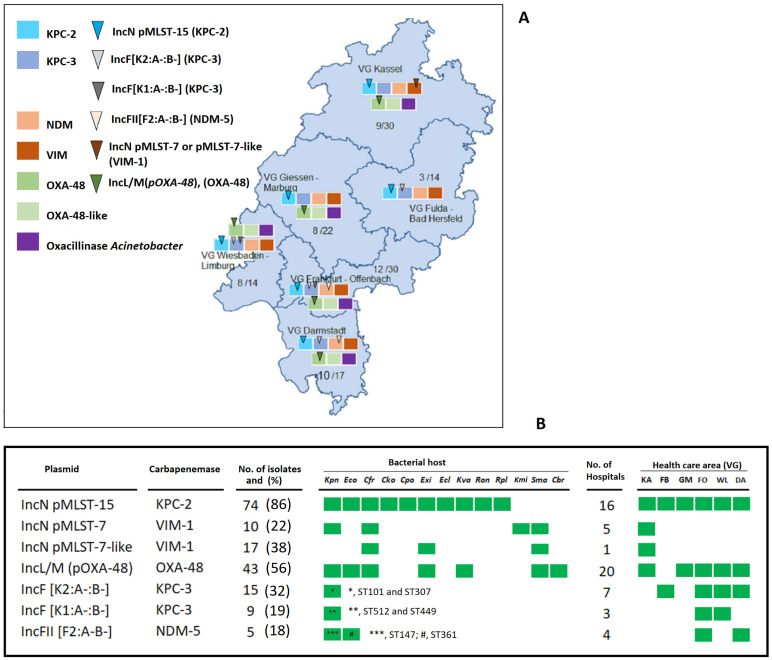
Characteristics of the most prevalent plasmids harboring KPC-2, KPC-3, VIM-1, NDM-5 and OXA-48 in Hesse, 2016–2019. (**A**) Geographical distribution; (**B**) Their occurrence in the bacterial species in Hesse from 2017 to 2019.

**Table 1 antibiotics-13-00682-t001:** Distribution of carbapenemases among carbapenem-non-susceptible Gram-negative bacterial species by year in Hesse, Germany, 2016–2019.

Carbapenemase	Species	Year				Total
		Before 2017	2017	2018	2019	
KPC	*K. pneumoniae*	18	28	11	13	70
	*E. coli*	3	2	6	10	21
	*C. freundii*	3	5	8	4	20
	*Enterobacter* spp.	–	1	4	4	9
	*C. portucalensis*	–	–	2	–	2
	*C. koseri*	–	–	–	2	2
	*K. aerogenes*	–	–	2	–	2
	*K. variicola*	–	1	2	–	2
	*Raoultella ornithinolytica*	–	–	1	–	1
	*Raoultella planticola*	–	–	2	–	2
	*All species*	24	37	38	33	132
NDM	*K. pneumoniae*	2	3	13	16	34
	*E. coli*	–	2	2	14	18
	*C. freundii*	–	–	1	2	3
	*Acinetobacter baumannii*	–	–	2	2	4
	*C. portucalensis*	–	–	–	1	1
	*Enterobacter* spp.	–	–	1	–	1
	*K. oxytoca*	–	–	–	1	1
	All species	2	5	19	36	62
VIM	*S. marcescens*	–	3	6	8	17
	*K. michiganensis*	–	6	3	2	11
	*C. freundii*	1	3	3	3	10
	*Pseudomonas aeruginosa*	–	3	4	–	7
	*K. pneumoniae*	–	5	–	1	6
	*Enterobacter* spp.	–	–	2	2	4
	*Proteus mirabilis*	–	–	–	3	3
	*C. portucalensis*	–	–	–	1	1
	*All species*	1	20	18	20	59
OXA-48 and	*K. pneumoniae*	–	11	31	19	61
OXA-48-like	*E. coli*	1	6	10	35	52
	*C. freundii*	–	1	6	8	15
	*S. marcescens*	–	2	–	4	6
	*Enterobacter* spp.	–	–	1	2	3
	*C. braakii*	–	–	–	1	1
	*K. michiganensis*	–	–	–	1	1
	*K. oxytoca*	–	1	–	–	1
	*K. variicola*	–	–	1	–	1
	*Raoultella ornithinolytica*	–	–	–	1	1
	All species	1	21	49	71	142
OXA-23	*A. baumannii*	1	15	7	13	36
OXA-72	*A. baumannii*	2	7	1	5	15
OXA-58	*A. baumannii*	–	–	–	1	1
GES-11	*A. baumannii*	–	–	1	–	1
SHV-38	*K. pneumoniae*	–	–	–	3	3

## Data Availability

The sequence data are available in NCBI’s GenBank under BioProject (Accession Numbers PRJNA602666 and PRJNA692829). The individual accession numbers for the raw sequence data are from SAMN13901101 to SAMN-13901119 in PRJNA602666 and SAMN17371722-17372171 as well as SAMN17372976 to SAMN17373145 in PRJNA692829.

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
