# Peer review of "Plasmid-Mediated Spread of Carbapenem Resistance in Enterobacterales: A Three-Year Genome-Based Survey"

_antibiotics, 2024, doi:10.3390/antibiotics13080682_

Round 1
Reviewer 1 Report
Comments and Suggestions for Authors
We commend the authors for their exemplary and insightful research demonstrated in their recent manuscript on the dissemination of carbapenemases among Enterobacterales. The study’s application of next-generation sequencing technology to trace and analyze the spread of carbapenem resistance through plasmids across multiple species and healthcare settings represents a significant advancement in our understanding of antimicrobial resistance mechanisms. This research is not only novel but also meticulously constructed, offering a high-resolution landscape of the resistance dynamics in a critical geographical area over a substantial period. The findings provide crucial information that could inform future surveillance strategies, antibiotic policy adjustments, and the development of targeted interventions to combat the spread of these resistant bacteria. The thorough analysis and clear presentation of complex genomic data underscore the importance of integrating genomic tools into public health efforts and enhance our capabilities in addressing one of the most pressing health threats of our time. We look forward to seeing the continued impact of this valuable work on both the scientific community and clinical practice.
However, there are some points to clarify, thus these comments aim to enhance the manuscript's scientific rigour, relevance, and readability, making it a valuable contribution to the field of antimicrobial resistance research.
Major comments:
1. Consider simplifying the title for broader accessibility. Perhaps, "Plasmid-Mediated Spread of Carbapenem Resistance in Enterobacterales: A Three-Year Genomic Survey".
2. Expand the abstract to include a brief mention of the methodology and key findings to immediately convey the unique contributions of the study. The introduction could benefit from a more detailed discussion of the clinical implications of carbapenem resistance.
3. Ensure that the literature review includes the most recent studies to contextualize the findings within current research, highlighting any new developments in carbapenemase detection and plasmid characterization.
4. Clarify the NGS technology used, including specifics about the sequencing platforms, read lengths, and assembly methods, to assist in reproducibility.
5. Enhance the clarity of the results section by structuring it around key themes, such as types of carbapenemases identified, distribution patterns, and linkage to plasmid types. Consider using subheadings to guide the reader through complex data.
6. Improve the resolution of Figure 1 and consider adding labels or a legend directly within the figure to improve readability. Ensure that all figures are high-resolution and suitable for publication.
7. Could you discuss any observed trends in the evolution or spread of specific carbapenemases over the three-year study period?
8. Expand the discussion to include potential mechanisms of plasmid transfer among different species and the implications for hospital hygiene and antibiotic stewardship. Discuss the limitations of the study, such as potential biases in hospital sampling and the geographic focus on Hesse, Germany.
9. Highlight how these findings can influence practical approaches in clinical settings, particularly in the development of strategies to mitigate the spread of resistance, such as targeted surveillance or modifications in antibiotic policies.
10. Suggest specific areas for future research, such as exploring the fitness costs associated with carrying resistance plasmids, or the role of other resistance mechanisms in the persistence and spread of carbapenem-resistant Enterobacterales. Consider calling for more extensive genomic studies across different geographical regions to validate findings and explore variations.
Comments on the Quality of English Language
Reviewer 2 Report
Comments and Suggestions for Authors
The present work broadly addresses interspecies dissemination mediated by carbapenemase plasmids in Enterobacterales. The data presented here are interesting and relevant since they can give an idea of ​​how the phenomenon of bacterial resistance is increasing. There are some points that can be improved. In the materials methods section, further describe the part of the statistical analysis package that was used. In the Whole Genome Sequencing section, although a citation is referenced, it will make it easier for the reader to give a brief description of the process. In the results part, the pathology and sex of the patients where the sample was obtained are described in tables; however, these data are not discussed or discussed in depth; it seems that they are not relevant, so it is important that they be discuss or are removed. In the discussion part we need to make more reference to other studies and compare them with what was found here.
Reviewer 3 Report
Comments and Suggestions for Authors
Summary
The authors conducted a genomic surveillance study of almost 600 CRNGB isolates sampled from 61 hospitals in Hesse, Germany, from 2017-2019. This is a massive amount of work that could have easily been put into 3 or 4 papers. It clearly adds to the local genomic epidemiology of CRNGB in Germany. The paper is well-written and structured. I just have a few comments that can enhance the paper.
Major comments
1. The authors want to run CheckM on all the genomes to ensure that they have decent values of completeness (>95%) and contamination (< 5%). Please do report this on the Supplementary Table 1.
2. Regarding the phylogenies, ideally, you want build them on core genome without recombination. Yet, it seems that some of them were run on the whole alignment generated by MAUVE. Also, please state which method (Maximum Likelihood, Bayesian, etc) was used for building the trees and their specifications (DNA substitution model, correction for invariable sites, etc.).
3. Some information must be added. In the introduction, a couple of sentences about the species prevalence in the region are desirable. The authors found some antibiotic resistance genes in different species (a hallmark of recent HGT) For instance, let’s say OXA-48. Maybe the authors want to say that HGT has played a role in the dissemination of antibiotic resistance in the discussion. Of note, this is in line with a recent study that has shown that HGT involving ARGs has occurred between Acinetobacter baumannii, Klebsiella pneumoniae and Escherichia coli (see ref below).
https://pubmed.ncbi.nlm.nih.gov/35075990/
4. Please add the hospital location for the isolates considering the phylogenies of the different species. This would be a useful way to see the dissemination across the hospitals.
Minor comments
Page 2, line 83: Please provide the actual references.
Page 13, line 305: you have two font sizes
The tree scale for the A. baumannii tree looks off
Page 17, Lines 463-466: maybe you want to compare this with these very recent publications about the global epidemiology of A. baumannii.
https://pubmed.ncbi.nlm.nih.gov/37882512/
https://pubmed.ncbi.nlm.nih.gov/37909743/
